# The Immunological and Hormonal Responses to Competitive Match-Play in Elite Soccer Players

**DOI:** 10.3390/ijerph191811784

**Published:** 2022-09-18

**Authors:** Ryland Morgans, Patrick Orme, Eduard Bezuglov, Rocco Di Michele, Alexandre Moreira

**Affiliations:** 1Department of Sports Medicine and Medical Rehabilitation, Sechenov State Medical University, 119991 Moscow, Russia; 2Sport Science and Medical Department, Bristol City FC, Bristol BS3 2EJ, UK; 3Department of Biomedical and Neuromotor Sciences, University of Bologna, 40126 Bologna, Italy; 4Department of Sport, School of Physical Education and Sport, University of Sao Paulo, Sao Paulo 05508-270, Brazil

**Keywords:** salivary cortisol, salivary immunoglobulin A, physical match performance, recovery, soccer

## Abstract

This study aimed to examine the salivary immunoglobulin A (s-IgA) and salivary cortisol (s-Cort) responses to competitive matches in elite male soccer players. Data were collected for 19 players (mean ± SD, age: 26 ± 4 years; weight: 80.5 ± 8.1 kg; height: 1.83 ± 0.07 m; body-fat 10.8 ± 0.7%) from a Russian Premier League team throughout a 6-week period during the 2021–2022 season. Physical match loads were measured through an optical tracking system. s-IgA and s-Cort were assessed one day before each match (MD − 1), 60-min before kick-off, 30-min post-match, and 48-h post-match (MD + 2). At 60-min before kick-off, s-IgA values were lower than at MD − 1 (90% CI difference 15.7–71.3 μg/mL). Additionally, compared to 60-min before kick-off, s-IgA was higher at 30-min post-match (90% CI difference 1.8–57.8 μg/mL) and at MD + 2 (90% CI difference 5.4–60.5 μg/mL). At 30-min post-match, s-Cort was higher than at 60-min before kick-off (90% CI difference 4.84–7.86 ng/mL), while on MD + 2 s-Cort was higher than at 60-min before kick-off (90% CI difference 0.76–3.72 ng/mL). Mixed model regressions revealed that longer playing time and total distance covered, and higher number of high-intensity accelerations, involved smaller s-IgA differences between 30-min post-match and 60-min before kick-off, and between 60-min before kick-off and MD + 2. Additionally, greater high-intensity and sprint distances, and a higher number of high-intensity and maximal accelerations, involved smaller s-Cort differences between 60-min before kick-off and MD + 2. In conclusion, the present results demonstrate that using salivary monitoring combined with match load may be a useful tool to monitor individual mucosal immunity and hormonal responses to match-play and the subsequent recovery periods in elite soccer players.

## 1. Introduction

The physiological demands of soccer performance have been extensively researched over the past several decades [1]. It is widely accepted that undertaking ~90 min of a soccer match induces significant disruption to bodily homeostatic parameters. The impact that this has on various physiological processes in the hours and days following match-play has also been researched in detail [2,3]. 

Various methods have been employed within research settings in an effort to quantify the physiological impact following soccer match-play. These methods include assessment of neuromuscular function [4], blood sampling [5], subjective questionnaires [6] and saliva sampling [7]. While these methods have been used effectively to highlight relationships between the physiological status of soccer players and training and match demands, there is a need to fully understand the profile of the response to elite competitive soccer match-play. This further understanding may allow practitioners to individualize the schedule and program of players to ensure full recovery following match-play, reducing the likelihood of injuries, and optimal physical preparation for upcoming matches to maximize subsequent performance. 

From the aforementioned methods, saliva sampling methods have been employed to quickly screen players for stress and illness on a regular basis throughout the season [7,8,9,10]. Saliva sampling is a relatively simple and non-invasive method that provides practitioners with a variety of markers that can be used to understand players physiological status pre- and post-match. Previous research has outlined the use of salivary markers such as salivary immunoglobulin A (s-IgA) [7], cortisol (s-Cort) [11], and testosterone [12] in soccer players following match-play.

As outlined above, the stressors of soccer match performance result in disruption to the physiological status of players. Mortatti et al. [8] reported a decrease in s-IgA concentration, a marker of mucosal immunity, in elite U19 soccer players when regularly monitored in a series of seven matches over 20 days, which may leave players more susceptible to illness, specifically through upper respiratory tract infections. Indeed, Springham et al. [10] also identified a cross-season suppression of s-IgA in professional soccer players, which was related to players perceived fatigue, sleep quality and muscle soreness suggesting the need to adopt s-IgA monitoring to aid in the prescription of training load and recovery. Therefore, methods that may be able to provide practitioners with an objective understanding of immune system function, in particular for mucosal immunity, in the period following match-play may be able to minimize the number of training days lost to illness over the course of a season [13]. 

Cortisol is a steroid hormone, detectable in saliva [14], that reflects catabolic balance [15]. Previous research has reported acute increases in s-Cort post-match in a variety of athletic populations including soccer [15,16], rugby [17], and Australian Rules football (AFL) [18], and differing training methods [19], which may persist for between 24- and 75-h [15,16,20]. Soccer studies that have examined longitudinal s-Cort responses have reported elevated values during periods of increased workload [21] and a reduction in Testosterone: Cortisol ratio toward the end of the competitive season [22]. However, previous longitudinal investigations are limited by infrequent or missing data points [21,22], while studies with short sampling periods have failed to examine the effect of elite competitive match-play or quantified the relationship between physical match performance and objective immunological (s-IgA) and hormonal (s-Cort) markers during the post-match 48-h recovery period. Thus, the ability to accurately analyze acute player responses is diminished. 

Morgans et al. [7] presented data that reported fluctuations in s-IgA to be sensitive to changes in the physical demands placed on soccer players as a result of changes in fixture scheduling at different time points across the season. Values for s-IgA were decreased during periods of condensed fixture schedules (2–3 matches per week) but returned to ‘normal’ baseline measures during regular fixture schedules (one match per week). Similar findings were presented by Mortatti et al. [8] when assessing changes in s-IgA during a period of congested fixtures (seven matches in 20 days). However, these authors found no change in s-Cort concentration during the same period. These authors also suggest that further investigation is required to better understand the potential relationship between s-Cort and the physical demands of elite soccer match-play. 

Therefore, this unique investigation aims to examine the s-IgA and s-Cort responses to match-play of elite European soccer players across six competitive fixtures compared with baseline and pre-match values, and to compare if and how these responses differ between starters and non-starters. Furthermore, the study aims to quantify the relationship between physical match performance and objective immunological (s-IgA) and hormonal (s-Cort) markers during the post-match 48-h recovery period. It was hypothesized that elite soccer match-play would induce changes in s-IgA and s-Cort when compared with baseline and that these changes would be greater for starters versus non-starters. 

## 2. Materials and Methods

### 2.1. Experimental Approach to the Problem

This study examined 19 elite male soccer players from the same team over a 6-week period during the second phase of the season. The participants had been playing soccer for a minimum of 10 years. Thirteen of the players used in this investigation were members of their respected national teams. The sample was initially recruited based on squad selection across six league matches (home matches (*n* = 4), away fixtures (*n* = 2)) in the 2021–2022 season. The sample was further sub-divided into starting players (*n* = 10) and non-starting players (*n* = 9). Participant data were only included in the analyses as starting player when time spent on the field exceeded 45-min of the match. Players were considered for inclusion as starting player if they completed, based on the inclusion criterion of 45-min playing time, in three (50%) or more of the examined matches. During a regular week, samples were obtained one day before each match (MD − 1), 60-min before kick-off on match-day, 30-min post-match and 48-h post-match (two days (MD + 2)). All samples were collected prior to breakfast in the morning period (09.30–10.30 a.m.) 1-h pre-training except on match-day. In the six examined matches, kick- off time was 2.00 p.m. (*n* = 3), 4.30 p.m. (*n* = 1), and 7.00 p.m. (*n* = 2). Sample collection time on match-day varied due to the official start of the match but was consistently 60-min prior to kick-off. In addition to saliva assessment, all match performance data was collated for analysis. Except on match-day, all participants were in a fasted state and required to abstain from food and caffeine products for a minimum of 2-h prior to the collection of saliva, and all salivary samples were collected at the same time of day for all participants (09.30–10.30 a.m.) to minimize the residual effect of exercise and circadian variations. 

### 2.2. Participants 

A total of 19 male outfield players (mean ± SD, age 26 ± 4 years; weight 80.5 ± 8.1 kg; height 1.83 ± 0.07 m; body-fat 10.8 ± 0.7%) were involved in the study. Players were classified by position and grouped accordingly: Center Defender (CD) *n* = 5, Wide Defender (WD) *n* = 3, Center Midfield (CM) *n* = 7, Wide Forward (WF) *n* = 2, and Center Forward (CF) *n* = 2. All data evolved as a result of employment in which players were routinely monitored over the course of the competitive season. Nevertheless, approval for the study from the club was obtained [23] and the study was performed in accordance with the Helsinki Declaration principles. Ethical approval was granted by the local Ethics Committee of Sechenov University (N 22-21 dated 12 December 2021). To ensure confidentiality, all data were anonymized before analysis. Participants were fully familiarized with the experimental procedures within this study due to the regular testing protocols implemented as part of the clubs’ performance monitoring strategy. During the study, players were instructed to maintain normal daily food and water intake, and no additional dietary interventions were undertaken. 

### 2.3. Procedures

The study period included saliva sampling and all match performance across a 6-week phase of the 2021–2022 season. The training sessions performed during the investigation were representative of a typical training micro-cycle implemented within elite European soccer, involving a periodized training week encompassing low, moderate, and high intensity sessions leading to competitive match-play. No player reported a soft tissue injury, illness or infection during the data collection period. 

### 2.4. Salivary Sampling

Given that soccer match-play induces a reduction in s-IgA concentration that return to basal levels within 18-h [24], we reasoned that collection of samples 48-h post-match would allow us to ascertain the effects of the acute suppression in s-IgA concentration from that associated with more chronic levels of stress. The diurnal rhythm of cortisol typically sees the highest concentrations in early morning with decreases as the day progresses [25]. Thus, players provided saliva samples pre-breakfast approximately 60-min before training on MD − 1, 60-min before kick-off on match-day, 30-min post-match and pre-breakfast approximately 60-min on MD + 2. 

Saliva samples were collected and analyzed from this cohort of players using the Soma OFC II collection kits in combination with real-time Lateral Flow Device (LFD), respectively. This method has been previously validated for oral fluid collection in the immunoassay of immunoglobulins in sports persons [26,27] and correlates well with other methods (enzyme-linked immunosorbent assay) adopted in the determination of s-IgA [9] and s-Cort [11,20,24]. In accordance with the manufacturer’s guidelines, after thoroughly rinsing their mouths with water, un-stimulated saliva samples were obtained. Players were required to place an Oral Fluid Collector (OFC II; Soma Bioscience, Oxfordshire, UK) consisting of a synthetic polymer-based swab attached to a polypropylene volume adequacy indicator stem in their mouth. Participants were instructed to swallow any saliva present within the oral cavity before placing the collection device on top of the tongue. Once the OFC kits collect 0.5 ml (± 20%) of oral fluid (collection time typically in the range of 20–50-s), the volume adequacy indicator turned blue and the player then placed the swab into the buffer bottle. The bottle was then mixed by gentle inversion for a period of 1–2-min, and the collected sample was ready to be analyzed through an IgA/Cortisol Dual LFD and photometric LFD reader (Soma Bioscience, Wallingford, UK). For the LFD, two-to-three drops of saliva/buffer mix were added to the sample window of the LFD cassette. The liquid in turn then ran the length of the test strip through creating a control and test line visible in the test window. Scanning of the LFD took place 15-min after the sample was added, being a competitive assay, the test line intensity was inversely proportional to the s-IgA and s-Cort concentration in the sample analyzed. This method has been previously validated [26,27,28] against ELISA (r^2^ = 0.78) in 208 samples collected from a cohort of English Premier League soccer players [28]. 

### 2.5. Physical Load 

League physical match performance data were collected using a two-camera optical tracking system (InStat, Moscow, Russia) that was installed to record and examine the technical and physical match performance during competitive league fixtures. The matches were filmed using two full HD, static cameras positioned on the centre line of the field, not less than 3-metres from the field and 7-metres in height. A consistent 25 Hz format was provided. Data were linearly interpolated to 50 Hz, smoothed using a 5-point moving average and then down-sampled to 10 Hz, which allowed analysis of all player actions with and without the ball [29]. The installation process, reliability, and validity of InStat have been previously reported [29]. Physical performance was analyzed using the InStat Analysis Software System and exported to the Microsoft Excel software for further analyses. InStat provided written permission to allow all match data to be used for research purposes. The physical match activity profile included: time on pitch (min); total distance covered (km); high intensity distance (km; total distance covered 5.5–7 m/s); sprint distance (km; total distance covered >7 m/s); number of high-intensity accelerations (peak speed 5.5–7 m/s); number of maximal accelerations (peak speed >7 m/s). 

### 2.6. Statistical Analysis

All data are presented as the mean ± SD. When appropriate, 90% confidence intervals (CI) were also shown. Data were analyzed with the software R, version 4.2.0 (R Foundation for Statistical Computing, Vienna, Austria). Linear mixed models, with random intercepts for individual players’ and match IDs, were used to assess the differences between the mean s-IgA and s-Cort values at the examined time points (MD − 1, 60-min before kick-off, 30-min post-match, and MD + 2) in starters compared to non-starters. The sample 60-min before kick-off was taken as the reference category to which values of MD − 1, 30-min post-match, and MD + 2 values were compared. Additionally, linear mixed-effect regressions with random intercept for players’ and match IDs were performed to examine the effect of playing time (min) and variables related to the match physical effort (distances covered and number of accelerations), on s-IgA and s-Cort, respectively. The s-IgA and s-Cort differences between post- (30-min after) and pre-match (60-min before), and the s-IgA and s-Cort differences between 48-h post- and pre-match, were taken as outcome variables. Effect sizes were calculated from the coefficients of linear mixed models as Cohen’s d through the lme.dscore function from the EMAtools package [30]. The absolute d value was interpreted as very small (<0.2), small (0.2–0.5), medium (0.5–0.8), large (>0.8). For all analyses, statistical significance was set at *p* < 0.10 due to the relatively small number of examined matches.

## 3. Results

The mean and SD s-IgA at the examined time points are shown in Figure 1. Sixty minutes before kick-off, the mean s-IgA value was significantly (*p* = 0.0108) lower than MD − 1, with an estimated difference of 43.5 μg/mL (90% CI: 15.7 to 71.3; d = 0.26, small). Additionally, compared to 60-min pre-match, there was a significantly higher value of s-IgA 30-min post-match (*p* = 0.083; estimated difference 29.8 μg/mL (90% CI: 1.8 to 57.8; d = 0.17, very small) and 48-h post-match (*p* = 0.051; estimated difference 33.0 μg/mL (90% CI: 5.4 to 60.5; d = 0.19, very small). No significant differences were observed between starters and non-starters at any time point, and there was no significant group x time interaction (*p* > 0.10).

Figure 2 shows the mean and SD values of s-Cort at the four examined time points. There was no significant difference between MD − 1 and 60-min before kick-off (*p* = 0.118). At 30-min post-match, s-Cort was significantly (*p* < 0.001) higher than 60-min pre-match, with an estimated difference of 6.35 ng/mL (90% CI: 4.84 to 7.86; d = 0.68, medium), while at 48-h post-match, s-Cort showed a decrease though it was still slightly higher (*p* = 0.014) than 60-min before kick-off, with an estimated difference of 2.47 ng/mL (90% CI: 0.76 to 3.72; d = 0.25 small). No differences were observed between starters and non-starters, and no significant time x group interaction was observed (*p* > 0.10).

Table 1 and Table 2 shows the coefficients of fixed effects obtained with linear mixed model analysis with playing time and physical match performance variables as fixed factors, and individual values of s-IgA differences, 30-min post-match vs. 60-min before kick-off (Table 1), and 48-h post-match vs. 60-min before kick-off (Table 2), as outcome variables. These coefficients indicate the change in s-IgA differences post-match involved by a one-unit increase of the independent variable in that given match.

A 1-min longer time on pitch involved a 0.74 μg/mL smaller 30-min post-match/60-min before kick-off difference, with a medium effect (Table 1), and a 1.32 μg/mL smaller 48-h post-match/60-min before kick-off s-IgA difference, with a medium effect (Table 2). Similarly, a greater total distance covered and a higher number of high-intensity accelerations involved smaller s-IgA differences between 60-min before kick-off and 30-min or 48-h post-match, with d values ranging from medium to large (Table 1 and Table 2). Additionally, greater high-intensity distance covered involved a smaller s-IgA difference between measurements taken 48-h post-match and 60-min before kick-off, with a medium effect (Table 2).

The fixed effects obtained from linear mixed models, with time on pitch and physical match performance variables as fixed factors, and individual values of s-Cort differences as outcome variables are presented in Table 3 (30-min post-match vs. 60-min before kick-off difference) and Table 4 (48-h post-match vs. 60-min before kick-off difference).

There was no significant effect of playing time, distances covered or the number of high-intensity or maximal accelerations on s-Cort differences between 30-min post-match and 60-min before kick-off (all *p* > 0.10) (Table 3). Conversely, greater high-intensity and sprint distances, and a higher number of high-intensity and maximal accelerations, involved smaller s-Cort differences between 48-h post-match and 60-min before kick-off, with small effects (Table 4). 

## 4. Discussion

This investigation aimed to examine the s-IgA and s-Cort responses to match-play of elite soccer players across six competitive fixtures in the 2021–2022 season compared with baseline and pre-match values. Furthermore, the study aimed to quantify the relationship between physical match performance and objective immunological (s-IgA) and hormonal (s-Cort) markers during the post-match 48-h recovery period. One of the main findings of the present study was the significant though slight decrease in s-IgA concentration from MD − 1 to 60-min before kick-off. It is reasonable to suggest that this result is somewhat unexpected as the release of s-IgA is under strong neuroendocrine control [31], and the activation of the sympathetic nervous system associated with player’s match preparation would, on the contrary, increase s-IgA concentration. Previously, it has been suggested that these mechanisms are responsible for the increases in s-IgA concentration induced by acute stress [32]. This result however, is unique in elite professional male soccer players and may suggest that psychological factors related to official match-play preparation may affect s-IgA concentration, and consequently, mucosal immune function. Moreira et al. [33], demonstrated in elite male volleyball players a significantly lower pre-match s-IgA concentration for a final championship match compared with pre-match s-IgA values for a regular season match. This result suggests that players’ perceived importance of the match affect s-IgA concentration, highlighting therefore, the role of psychological factors in modulating the mucosal immunity in team-sport athletes. Indeed, this result further indicates that monitoring resting s-IgA in team-sports athletes would provide valuable information regarding how athletes cope with competition induced stress. 

Regarding coping with stress related to competitive match preparation, the present results reported lower s-IgA concentration 60-min before kick-off compared to MD − 1, which may be partly explained by the well-known differences in responses to acute stress between active and passive coping strategies [34]. Bosch et al. [34] examined the acute immunological effects of two different laboratory stressors (“active coping” via a time-paced memory test and “passive coping” via a stressful video showing surgical operations). The results of the study showed that active coping led to increases in s-IgA concentration, while, passive coping induced a decrease in s-IgA concentration. Considering that the preparation for an official match may impose a significant psychological stress on team-sports athletes [33,35], it may therefore be inferred that the adoption of passive coping strategies before official match-play may negatively impact the mucosal immune function which in turn may increase the likelihood of upper respiratory tract infection occurrences. The present results in conjunction with the aforementioned data may possibly provide an opportunity for sport scientists and professionals working with soccer players to adopt active coping strategies during the preparation period for official soccer matches, and highlight the potential for the introduction of affective or positive emotional engagement. Further studies should focus on examining whether structured active coping tasks minimize the negative effect (i.e., decreasing s-IgA concentration) of the inherent stress associated with preparation for official match-play.

The current results also demonstrated an increasing trend in s-IgA concentration at 30-min and 48-h post-match, compared to 60-min before kick-off. These results suggest a short-term (acute) stress response induced enhancement of mucosal immune function [36]. Psychological and physiological stressors have been shown to stimulate biological stress. These responses are signals to cells and tissues, which express themselves as receptors for the released biological factors, leading therefore to the activation of all bodily systems, including the immune system. In contrast to chronic stress, that may lead to suppression or dysregulation of immune function, while impacting negatively the mucosal immunity [37], the present results suggest that the short-term stressors related to official soccer match-play may induce enhancement of immune function in professional soccer players. This is a positive response which prepares athletes for the imposed challenges associated with competition. It is important to highlight that previous studies have shown that factors such as corticosterone and epinephrine, released due to the presence of a stressor, are mediators of a short-term stress induced immuno-enhancement, while a variety of studies have shown increases or no changes in s-IgA concentration from pre- to post-match in team-sport athletes [9,33,38], professional female soccer players [39], and professional male soccer players [40]. Previous studies in soccer players demonstrated that elevated levels of psycho-physiological stress may negatively affect the mucosal immune function, with decreases in s-IgA concentration across periods of congested fixtures or intensive training loads [7,8,12,41]. Considering our results in combination with the existing literature, it could be reasonable to suggest that the probability of observing no changes or even increases in s-IgA concentration is high for acute stress (i.e., from pre- to post-match), while on the other hand, the chronic effect of accumulated stress, notably, when performing successive matches in a short period of time, may negatively affect the mucosal immunity of players. 

The design of the present study allowed the observation of s-IgA responses to actual physical match load that have not yet been demonstrated in official soccer matches with elite professional male players. Despite the observed trend to increase s-IgA concentration from pre-match to 30-min and 48-h post-match, it is notable that, when performing a higher workload, players seemed to present a slower return to their initial s-IgA concentration. The 1-min longer playing time on pitch produced a 0.74 μg/mL smaller 30-min post-match/60-min before kick-off difference and a 1.32 μg/mL smaller 48-h post-match/60-min before kick-off s-IgA difference. Smaller s-IgA concentration differences between 60-min before kick-off and 30-min or 48-h post-match were also observed in association with greater total distance covered, and with a higher number of high-intensity accelerations. Additionally, greater high-intensity distance covered involved a smaller s-IgA difference between 48-h post-match and 60-min before kick-off. This unique and important finding of the present study suggests that an inverted-U/bell-shaped relationship may be observed between match-workload and the effects on mucosal immune function. Thus, when performing higher workload, above a given threshold, players may be more prone to trivial increases or even reductions in s-IgA concentrations. In addition, this result may aid in explaining the increased likelihood of a suppressed effect from accumulated and successive match-play in s-IgA concentration, as this workload accumulation would affect plasma cells functions (immunoglobulin-secreting plasma cells) and the rate of IgA transcytosis across the epithelial cell. This result suggests a novel role for physical match workload monitoring and its impact on mucosal immunity in professional soccer players. 

In relation to s-Cort, there was no significant difference between MD − 1 and 60-min before kick-off. This result suggests that the expected anticipatory stress response to match participation [42] did not occur. This finding might be associated with the high-level of the examined players and with their habitual lead-in process to cope with the pressure and anxiety involved in the period preceding the start of official matches. In this sense, van Paridon et al. [42] reported in their systematic review that the anticipatory stress response and cortisol reactivity, in both male and female athletes competing at international level, do not present a significant anticipatory cortisol response. Moreover, in earlier research, Alix-Sy et al. [43] despite showing a significant increase in s-Cort concentration at pre-match compared to a non-training day in professional French soccer players, reported a significant positive association between unpleasant somatic emotions and cortisol. Indeed, Alix-Sy et al. [43] also demonstrated no differences in s-Cort between starters and non-starters, as observed in the current study. Furthermore, it should be highlighted that in their study, the authors compared a non-training day with official matches, while in the present study, saliva collection occurred during habitual training sessions performed one day before matches. This difference may influence, at least in part, the present result of no change in s-Cort.

Considering these findings, we might suggest that the players evaluated in the present study did not show a s-Cort rise from MD − 1 to 60-min before kick-off possibly due to their positive evaluation of the potential match challenges, which in turn may be related to their perception of relative situational control and the non-decisive nature of regular season matches. Considering the results of the present study it may be suggested that due to the nature of the evaluated matches and the level of the assessed players, the s-Cort anticipatory responses (MD − 1 vs. 60-min before kick-off) indicated an optimal cognitive and behavioural player state to participate in the matches. 

As expected, a significant increase in s-Cort from 60-min before kick-off to 30-min post-match was observed, while at 48-h post-match, s-Cort showed a decrease though still slightly higher than 60-min before kick-off, and no differences were observed between starters and non-starters. The increases in s-Cort reinforces that official soccer match-play induce significant psychophysiological stress likely related to physical demands associated with the volume and intensity of match-play, leading to increased secretion of s-Cort, as also reported in A-League [16] and intercollegiate soccer players [44]. It is important to highlight that the psychological factors involved in official match-play may play a role in this result. These results in conjunction allow us to infer that besides the well-known effect of increased s-Cort related to exercise stress, which represents per se a potent physiological stressor [45], the pressure of official match-play may be considered as an additional stress factor, possibly due to its social-evaluative task characteristics combined with other contextual factors inherent to sports competition as proposed by Arruda et al. [46].

Indeed, as demonstrated more recently by Rowell et al. [16], a substantial individual variability in s-Cort response to soccer match-play may be expected, including the responses within 48-h post-match. Furthermore, the psycho-physiological relationships and the impact of situational factors have been reported to influence cortisol responses to match-play in soccer players [47]. Thus, the present results add to the literature and suggest that contextual factors other than being a starter or non-starter may influence the variability in players s-Cort responses. The uniqueness of the present study allowed us to examine the effect of match-load measures on s-Cort time-course responses. A novel and interesting finding of the present study was that the greater high-intensity distance, sprint distance, number of high-intensity and maximal accelerations performed, the smaller the s-Cort differences between 48-h post-match and 60-min before kick-off. This result suggests that performing a greater number of high-intensity actions during the match would increase the associated stress, that in turn may hinder s-Cort recovery to resting values. Moreover, the present results suggest that high-intensity distance, sprint distance, number of high-intensity and maximal accelerations may be employed as reliable markers of individual external match-load inducing stress, and possibly predict catabolic state induced by match-play, rather than dividing players into starters and non-starters groups. In addition, the findings highlight the need to monitor in conjunction with the individual external match-load and the s-Cort response of players to account for individual variability in recovery from match-play. The results also suggest that examining s-Cort responses from pre-match to 30-min post-match might not aid in observing true changes in s-Cort during the recovery time-course. 

Despite the interesting findings of the current study, some limitations should be acknowledged. Firstly, our study only focused on one elite European professional soccer team across a 6-week period, and as a result, the findings and practical implications must be considered with caution when applying to another set of players from a league with different characteristics such as match demands, travel [48], climate [49], and over an extended period of time during a different phase of the season (early-, mid-, late-season, congested Christmas schedule). Furthermore, the sample size was also a limitation due to this study being conducted in the real-world, conducted with players from an elite soccer club. Our sample was selected as a convenience sample by recruiting all available outfield players from the first team of the club involved. Nevertheless, similar sample sizes have been used in previous studies conducted in elite soccer players in this research field. Secondly, match outcome was not considered, which has the potential to affect immunological and hormonal recovery profiles. Future investigations are warranted to evaluate these factors as they may be particularly relevant in different leagues across varying athletic populations during the season. Other limitations include the absence of training load, fatigue, and fitness profiling data [50]. 

### Practical Implications

The present findings may provide practitioners with detailed knowledge about acute and chronic variations in physical match performance and the subsequent recovery responses, that can be practically useful to assess and interpret change in individual and team performance. Previously, a number of practical recommendations to monitor immune function in athletes have been documented [7,9,10,16,51]. Match-play with higher physical outputs did not necessarily produce disturbances to mucosal immunity and hormonal balance. Therefore, accordingly, designing a structured, planned and individualized tailored recovery strategy and potential for squad rotation should be considered during demanding stages of the season to ensure immunological and hormonal recovery. Previous results highlighted that this might be particularly important during congested fixture schedules (Christmas fixture period) [7] and toward the end of the season [10]. Our findings support the use of s-IgA and s-Cort monitoring in professional soccer players and devising individual thresholds to determine values associated with inadequate recovery. 

## 5. Conclusions

As a result of this specific investigation, the data demonstrate for the first time that the use of salivary monitoring in combination with physical match load may be a useful tool to monitor individual mucosal immunity and hormonal responses to elite soccer match-play and the subsequent recovery periods. However, surprisingly no significant differences were observed between starters and non-starters at any time point, thus additional research is required. Finally, analysis of specific time points during recovery also warrants further investigation. 

## Figures and Tables

**Figure 1 ijerph-19-11784-f001:**
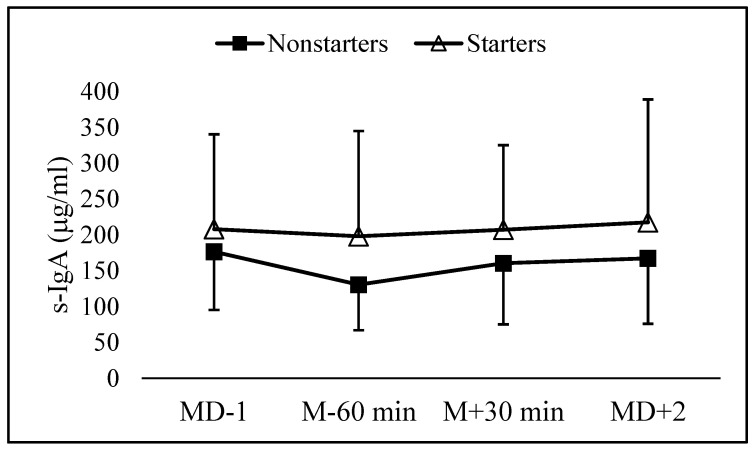
Mean and SD values of s-IgA the day before the match (MD − 1), 60-min before kick-off (M − 60 min), 30-min post-match (M + 30 min), and 48-h post-match (MD + 2).

**Figure 2 ijerph-19-11784-f002:**
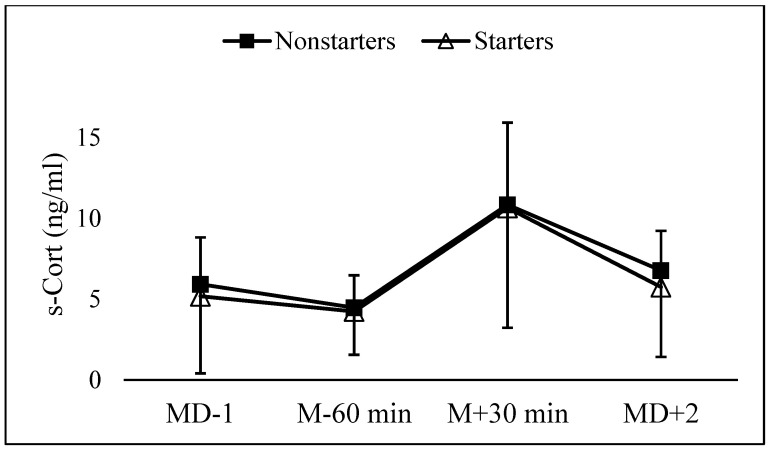
Mean and SD values of s-Cort the day before the match (MD − 1), 60-min before kick-off (M − 60 min), 30-min post-match (M + 30 min), and 48-h post-match (MD + 2).

**Table 1 ijerph-19-11784-t001:** Effects of playing time and physical performance on s-IgA differences calculated between 30-min post-match and 60-min before kick-off time points.

	Coefficient (90% CI)	*p*-Value	Cohen’s d
Playing time (min)	−0.74 (−1.36 to −0.12)	0.059 *	0.62
TD (km)	−6.82 (−12.36 to −1.31)	0.051 *	0.61
High-intensity distance (km)	−64.91 (−128.34 to −1.16)	0.102	0.43
Sprint distance (km)	61.99 (−200.82 to 320.77)	0.697	0.09
Number of high-intensity accelerations	−1.18 (−2.16 to 0.19)	0.057 *	0.48
Number of maximal accelerations	−0.42 (−4.89 to 3.98)	0.876	0.03

* *p* < 0.10. TD = Total distance; CI: Confidence Interval.

**Table 2 ijerph-19-11784-t002:** Effects of playing time and physical performance on s-IgA differences calculated between 48-h post-match and 60-min before kick-off time points.

	Coefficient (90% CI)	*p*-Value	Cohen’s d
Playing time (min)	−1.32 (−2.18 to −0.45)	0.013 *	0.80
TD (km)	−12.61 (−20.20 to −4.95)	0.007 *	0.81
High-intensity distance (km)	−125.65 (−211.21 to −37.23)	0.018 *	0.61
Sprint distance (km)	−323.09 (−679.52 to 49.44)	0.145	0.33
Number of high-intensity accelerations	−1.71 (−3.03 to −0.35)	0.037 *	0.52
Number of maximal accelerations	−3.16 (−9.20 to 3.18)	0.396	0.19

* *p* < 0.10. TD = Total distance; CI: Confidence Interval.

**Table 3 ijerph-19-11784-t003:** Effects of playing time and physical performance on s-Cort differences calculated between 30-min post-match and 60-min before kick-off time points.

	Coefficient (90% CI)	*p*-Value	Cohen’s d
Playing time (min)	0.005 (−0.042 to 0.053)	0.838	0.06
TD (km)	0.01 (−0.41 to 0.42)	0.999	0.00
High-intensity distance (km)	0.31 (−4.24 to 4.83)	0.910	0.03
Sprint distance (km)	1.09 (−17.42 to 18.78)	0.921	0.02
Number of high-intensity accelerations	−0.011 (−0.081 to 0.059)	0.778	0.07
Number of maximal accelerations	0.073 (−0.243 to 0.371)	0.686	0.09

TD = Total distance; CI: Confidence Interval.

**Table 4 ijerph-19-11784-t004:** Effects of playing time and physical performance on s-Cort differences calculated between 48-h post-match and 60-min before kick-off time points.

	Coefficient (90% CI)	*p*-Value	Cohen’s d
Playing time (min)	−0.029 (−0.065 to 0.007)	0.193	0.35
TD (km)	−0.30 (−0.62 to 0.01)	0.116	0.41
High-intensity distance (km)	−3.54 (−7.02 to −0.11)	0.097 *	0.38
Sprint distance (km)	−15.55 (−29.06 to −2.06)	0.063 *	0.43
Number of high-intensity accelerations	−0.0056 (−0.1098 to −0.0035)	0.086 *	0.40
Number of maximal accelerations	−0.236 (−0.462 to −0.009)	0.091 *	0.39

* *p* < 0.10. TD = Total distance; CI: Confidence Interval.

## Data Availability

The data are not publicly available due to privacy reasons.

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
