# Peer review of "The Immunological and Hormonal Responses to Competitive Match-Play in Elite Soccer Players"

_ijerph, 2022, doi:10.3390/ijerph191811784_

Round 1

Author Response

Reviewer 1

This study is mostly about the immunological and hormonal responses to competitive match-play in elite soccer players. The article is well written and well designed but I have some minör corrections.

We thank the Reviewer for his/her appreciation of our work. We have modified the relevant parts of the manuscript according to the Reviewer’s comments.

  • The introduction is good and leaves the reader to the main point of the study. But the last sentence of the first paragraph (Line – 39-42) can be better to move at the end of paragraph or deleted from here.

RE: We agree that this sentence was out of context in the paragraph, so we deleted it.

  • Please give detailed information about the hypothesis of the study. For example, the authors did not mention starters and nonstarters comparison...

RE: In the aims the starters vs. non starters comparison is mentioned (L97-100). Additionally, we have now stated a hypothesis (L102-105).

  • In the experimental approach to the problem section, what are the other inclusion and exclusion criteria of the study? For example, the authors give attention to the use of medication on match day. Also, some of the inclusion criteria are explained bellowed sections but if the authors add to them here it will be better to understand the experimental approach.

RE: We have now reported all inclusion criteria in the “experimental approach to the problem” section.

  • The authors need to add the starting times of 6 official games (lines 110-111) than say “…. collection time on match-day varied due to the official start of the match”.

RE: amended

  • In the Physical Load section, Line 191-194, please add the appropriate reference for this statement

RE: amended in reference list and in text

  • The authors did not present a sample size calculation. Did you perform power analysis to find the sample size?

RE: we didn’t perform a power analysis since we used a convenience sample constituted by all available outfield players from the 1st team of the elite club involved with the study (n=19). Similar sample sizes have been used in previous studies conducted on elite soccer players in this research field and we consider this sample size to warrant enough statistical power for the statistical analysis carried out.

  • The results are clear and well presented, just add the abbreviation part under the tables.

RE: amended

  • The limitation can be expanded. Despite the mentioned limitation, the sample size can be a major limitation of the study

RE: inserted (L443-447)

  • The authors can write better conclusion statements when considering the findings and limitations of the study. For example, I didn’t see any statement about the other aim of the study that compares the starters and nonstarters.

RE: inserted (L473-475)

Reviewer 2 Report

The manuscript is very well designed and very well writen.

shows clarity and results evidence.

the literature support could be improved and the only wickness in the manuscript is the number of participants (19)...

Author Response

Reviewer 2

The manuscript is very well designed and very well written.

shows clarity and results evidence.

the literature support could be improved and the only wickness in the manuscript is the number of participants (19)...

We thank the Reviewer for his/her kind opinion about our work. We agree on the fact that the number of participants is not extremely large, but as we also pointed out in response to a comment about power analysis from Reviewer # 1, being this a real-world field study conducted with players form an elite club, our sample was selected as a convenience sample by recruiting all available outfield players from the 1st team of the club involved. Nevertheless, similar sample sizes have been used in previous studies conducted on elite soccer players in this research field and we consider this sample size to be large enough for the statistical analysis that was carried out.

Reviewer 3 Report

I am grateful to the editor for giving me the opportunity to review this manuscript. The aim of the study was to examine the salivary immunoglobulin A and salivary cortisol responses to competitive matches in elite male soccer players. The article is well developed and presents interesting results. There are some aspects that need to be improved before publication.

Abstract

Authors are encouraged to shorten the abstract. In my opinion it is too long.

Introduction

The introduction correctly presents the study question. However, there are certain references that are outdated (e.g. number 11). It is advisable to replace them with more up-to-date ones.

Materials and methods

The method is correctly presented.

Results

The abbreviations used in the tables and figures should be explained in the table captions.

Authors are advised not to repeat information in tables and text.

Discussion

In general, the discussion is well structured and well developed.

Some references (e.g. 47) are outdated and should be replaced by more current ones.

Author Response

Reviewer 3

I am grateful to the editor for giving me the opportunity to review this manuscript. The aim of the study was to examine the salivary immunoglobulin A and salivary cortisol responses to competitive matches in elite male soccer players. The article is well developed and presents interesting results. There are some aspects that need to be improved before publication.

We thank the Reviewer for his/her positive assessment and his helpful comments. We have modified the relevant parts of the manuscript according to the Reviewer’s comments.

Abstract

Authors are encouraged to shorten the abstract. In my opinion it is too long.

RE: We had to write a quite long abstract to include all the results we believe should be reported in the abstract. Nevertheless, we have now deleted a few unnecessary words so the abstract is slightly shortened.

Introduction

The introduction correctly presents the study question. However, there are certain references that are outdated (e.g. number 11). It is advisable to replace them with more up-to-date ones.

RE: amended in Reference list

Materials and methods

The method is correctly presented.

Results

The abbreviations used in the tables and figures should be explained in the table captions.

RE: amended

Authors are advised not to repeat information in tables and text.

RE: We removed from the text the effect sizes values, already presented in the tables

Discussion

In general, the discussion is well structured and well developed.

Some references (e.g. 47) are outdated and should be replaced by more current ones.

RE: amended in text (L417-426) and reference list

We thank the Reviewers for their thoughtful comments. We hope to have addressed all your concerns.